# The 4P: Preventing Preneoplasia through Patients Partnership

**DOI:** 10.3390/cancers13174408

**Published:** 2021-08-31

**Authors:** Angélique Virgone, Sara Badreh

**Affiliations:** 1Tumor Escape, Resistance and Immunity Department, Université Lyon, Université Claude Bernard Lyon 1, INSERM 1052, CNRS 5286, Centre Léon Bérard, Centre De Recherche En Cancérologie De Lyon, 69008 Lyon, France; 2Centre Léon Bérard, Department of Translational Medicine, 69008 Lyon, France; 3Secretariat Department of the European Cancer Patient Coalition (ECPC), 1000 Bruxelles, Belgium; sara.badreh@ecpc.org

**Keywords:** prevention, patients, oral cancers, preneoplasia, partnership

## Abstract

**Simple Summary:**

This commentary article is a cross-examination between the research and the associative environments around the cancers of the oral cavity. The authors address the question of a better prevention of oral potentially malignant disorders. They aim to outline the actions that could be deployed by patient advocacy groups to successfully complete oral potentially malignant disorders prevention challenge. They explain that there is a clear added value of researchers and patient representatives working together to best approach early diagnosis and improve education of disease signs and symptoms by awareness campaigns in addition to the existing, for effective prevention since the beginning.

**Abstract:**

The early diagnosis and management of oral potentially malignant disorders (OPMD) represent a unique opportunity to develop strategies that will prevent malignant transformation. Despite a high prevalence, awareness remains low, patient outcomes poor, and quality of life highly affected. How can patient advocacy groups (PAGs) bring more awareness to preneoplasia preceding oral cancers and help patients after the identification of a suspicious oral leukoplakia presented as white patches in the mouth? PAGs are today involved with awareness campaigns, lobbying, and education of both health care systems as well as the survivor and the newly diagnosed. PAGs are a link between the clinician and the patient, making sure that the medical terminology used is explained in layman language and that psychological support is available during and after treatment. This review outlines the actions that could be deployed by PAGs to successfully complete OPMD prevention challenge. The added value of researchers and patient representatives working together is the increased awareness of the problem. To know at which angle to best approach it for encouraging early diagnosis, improved education of disease signs and symptoms will condition effective prevention from the beginning.

## 1. Introduction

World Health Organization (WHO) classification defines oral potentially malignant disorders (OPMD) as “clinical presentations that carry a risk of cancer development in the oral cavity, whether in clinically definable precursor lesions or in clinically normal oral mucosa” [1]. The oral cavity is the most common site of head and neck squamous cell carcinoma (HNSCC). Worldwide, around 350,000 new patients are diagnosed with oral cavity squamous cell carcinoma (OCSCC) every year, with a mortality rate of 175,000 per year and 5-year overall survival is about 50% (Figure 1). Regional variations in incidence and the site of occurrence relate to the major causes of alcohol and smoking in Western countries and betel quid and tobacco chewing in South and Southeast Asia [2]. Around 15% of the patients with OCSCC are non-smokers non-drinkers with no clear link with the human papillomavirus infection has been found. OPMD estimated global prevalence is 4.47% in a recent meta-analysis, and the diagnosis of OCSCC is often made at a late stage. Despite these facts and numbers, awareness remains low, patient outcomes remain poor, and quality of life is highly affected. Oral cancer has a long preclinical phase that consists of the appearances of precancerous lesions. In western countries, oral leukoplakia (presented as white patches) is the most prevalent sign of precancerous lesions, affecting males more frequently, and tobacco use is the most frequent cause. Among the precancerous lesions, we find homogeneous or nonhomogeneous leukoplakia, verrucous leukoplakia, erythroplakia, oral submucous fibrosis, lichen planus, and chronic traumatic ulcers [3]. The estimated annual frequency of malignant transformation of OPMD to OCSCC ranges from 0.13 % to 2.2 % [4]. OPMD is clinically detectable through visual inspection, and palpation of the oral mucosa, and lower than four centimeters lesions can be resected surgically or treated with radiotherapy, raising the 5-year survival rate to 80%. A meta-analysis showed an estimated mean for malignant transformation of 12%, ranging from 0% to 36%. Therefore, the main challenges for the prevention of OPMD transformation into cancer are to identify high-risk patients and treat them with chemopreventive therapies in order to prevent OPMD transformation of the entire oral mucosa [5].

Besides, OPMD shows a high rate of relapse. The proportion of OCSCC preceded by OPMD is largely unknown, and that is the major difficulty of this precancer. The early diagnosis and management of OPMD represent a unique opportunity to develop strategies that will prevent malignant transformation, in addition to the existing such as avoiding tobacco/alcohol exposures. A multiple approach integrating health education, tobacco/alcohol decrease, and early detection is needed to intercept OPMD to OCSCC transformation [4,6]. Improving awareness of patients and caregivers and better screening of these lesions at the earlier stages are the key measures to slow down oral cancers occurrence. Patient advocacy groups (PAGs) can be at the origin of this initiative. This review outlines the actions that could be deployed by PAGs to successfully complete OPMD prevention challenge. There is a clear added value of researchers and patient representatives working together to best approach early diagnosis, improve the education of disease symptoms by awareness campaigns, in addition to the existing, for effective prevention since the beginning.

## 2. Three Prevention Degrees

Current treatment and rehabilitation of patients with OCSCC cause a high economic burden, which is likely increased further with the latest treatment developments (immunotherapy and advanced radiotherapy). In this context, the prevention of cancer or its recurrence becomes increasingly important. The issue of prevention is twofold: human and economic. In Europe, the average medical cost per patient with oral cancer in 2012 represented between 20,000 and 23,000 euros, and the average hospital stay was around 2 weeks [7].

World Health Organization (WHO) distinguishes 3 types of prevention [8]:

Primary prevention acts upstream of the disease (e.g., vaccination and action on risk factors). It aims at increasing public awareness and changes the general public behavior regarding the risk factors. Despite a significant decrease in smoking and alcohol abuse, the proportion of smokers still ranges from 9 to 28%. Some awareness campaigns exist for HN cancers (See Section 4) as the awareness of the disease is alarmingly low.

Secondary prevention acts at an early stage of cancer development and aims at early detection of OPMD and OCSCC through clinical screening. No public awareness campaigns are currently focused on the diagnosis of OPMD, and no efficient screening programs exist for HNSCC in Europe. Therefore, there is an urgent need not only to raise awareness of the signs and symptoms of the disease but also to educate the general public and healthcare providers on the importance of prevention and regular screening.

Tertiary prevention aims at preventing the development of second primary tumors. Tertiary prevention is indicated when a patient has previously developed oral cancer and acts on minimizing morbidity and the risk of recurrence. 

Regular monitoring (follow-up) of patients with oral cancer and precancer (secondary prevention) forms the basis of tertiary prevention [9].

This classical view of prevention is limited by the fact that it is segmented and not patient-centered. Innovative patient-centered and precision preventive strategies are, therefore, needed [10,11]. However, prevention programs need a certain amount of upfront investment and because uncertain outcomes may lead to overdiagnosis, overtreatment, and waste of resources.

The development of preventive medicine has important sociological and ethical implications. Patient autonomy vs. wellbeing ethical approaches can be discussed considering the fact that patients with OPMD are often almost asymptomatic and may never develop cancer. In the “cure” versus “care“ ethical approaches, precision preventive medicine will require biospecimens for patient stratification. 

### 2.1. Primary Prevention: No Smoking/No Drinking Campaigns

Regarding primary prevention, the first major strategy is to reduce access to tobacco by increasing the price of tobacco and reduce advertising promoting the use of cigarettes and alcohol. In the case of tobacco cessation, increasing the cost of tobacco is the most cost-effective approach [12]. 

### 2.2. Secondary Prevention: Catching Cancers Early

#### 2.2.1. Visual Oral Cancer Screening

A variety of Health Care Professionals (HCPro), including dentists, general practitioners, oncologists, surgeons, nurses, but also caregivers, may provide oral visual screening after training. 

The first step to detect OPMD is the medical examination, in which dentists look for signs, changes in the oral mucosa, ask patients about chicha, betel quid, cannabis consumption, and look for HPV status. The warning signs for the practitioner could be an increasing evolution of previous signs, bleeding, or dysphagia. The examination of the oral mucosa is painless, and the detection is conducted via lights chemiluminescence exposure system that makes the lesions appear white, mirrors, and compresses. Any lesion is palpated, in particular, around lymph node areas, and in case of abnormalities, the evaluation by an ear nose and throat (ENT) specialist is highly required. Dentists usually use cytobrushes (Orcellex^®^, Rovers Medical Devices, Oss, The Netherlands) that allow for spreading the sample on a slide and addressing it to a laboratory. This method is proposed as a highly accurate one for detecting OPMD.

Dentists have a major pre-screening role [13]. They are missioned by the dental college to perform a clinical examination with the recommendation to biopsy or refer to a surgeon or oncologist. These precancerous lesions must be systematically biopsied with an appropriate follow-up. The major problems emerging from this primary way of prevention by dentists may be that they do not do enough biopsy either out of fear (illegitimacy feeling), lack of material, and lack of Continuing Professional Education or initial training. In any case, the current watchword is to involve them actively in this early detection and address patients in case of doubt. Earlier detection of OPMD must become part of the routine examination. It has been described that an annual oral examination carried out by a primary care dentist can detect mucosa abnormalities [1,13,14,15].

The role of the pharmacists’ community is also fundamental in OPMD screening, although this system may be different depending on the country. In France, for example, pharmacists are the first line of information about tobacco, alcohol, or cannabis (information brochures, dialogue, fight against preconceived ideas) and help with quitting tobacco and alcohol (patch, oral drugs on medical prescription or not, electronic cigarettes).

Pharmacists have a reorientation role to doctors, tobacco specialists, addictology centers, nutritionists, but also encourage participation in awareness campaigns to bring patients to a reflection. Even for prevention measures (HPV vaccination or other campaigns), the pharmacist has an important role. The pharmacist has a privileged link with the patient that may be followed for related diseases and that they are the first to be able to guide them in their care trajectory.

Sensitivity ranges from 40% to 93%, and specificity ranges from 50% to 99% for detecting OPMD [4,16,17]. The American Dental Association (ADA) expert panel review on population-based oral cancer screening [18] recommended a routine examination of the intraoral mucosa and lymph nodes under bright light screening in all patients particularly in tobacco and/or alcohol users, that can reduce the occurrence of OCSCC [18,19]. Although this test is subjective, its performance in detecting lesions differs between individuals. In the most serious cases, the consequences of an oral screening are incisional or excisional biopsies. In this type of screening, it is important to take into account geographic gaps in the availability of cancer diagnostic [20]. Some developed countries already follow this screening routine for populations over 30 years of age that have a risk of developing oral cancers, but for example, in Southeast Asian countries, the resources of all countries do not necessarily allow this visual oral screening even if it is a national priority [21,22]. Europe involvement in the detection of these diseases must be concretized by a systematic examination of the oral cavity. In the United States and the United Kingdom, 82 to 84% of dental surgeons already perform such screening. Dentists must be aware that these five minutes of examination can save a life [23].

#### 2.2.2. Self-Examination and Other Screening Methods

Mouth self-examination using a mirror was evaluated as an effective screening test in high-risk sites: the floor of the mouth, the ventrolateral surface of the tongue, and the soft palate [24]. 

Current methods lie in the use of screening methods such as toluidine blue staining (Vizilite Plus^®^, DenMat, Lompoc, CA 93436, USA), autofluorescence (VELscope^®^, LED Dental, Vancouver, BC V6C 3B6, Canada), chemiluminescence and salivary analysis [25,26,27,28] to make fluorescent in the mouth any invisible asymptomatic lesions. In order to set up this type of self-examinations screening, it could initially address on a smaller scale to people over 40 who are smokers/drinkers.

### 2.3. Tertiary Prevention: No Tumoral Relapse

Dentists are again on the front line when it comes to monitoring their patients on whom they have already performed a treatment. It is essential to strengthen the links between patients and caregivers in order to ensure this continuity of care. Dentists are also good at keeping strong interprofessional links with the surgeons to whom the patients have been referred to ensure effective follow-up and avoid tumoral relapse. Up to 15% might relapse and develop a second primary tumor. Continuing dental care by annual examinations in the first five years is important [2] even if the geographical factors and the social level intervene in this continuity of care. In order to reduce the need for visits to a specialized center, patients have to be helped in this way. 

## 3. Future Needs

### 3.1. The Help of «First Line» Practitioners: Dentists/Dental Hygienists and Pharmacists

As previously mentioned, dentists have a major role in preventing OPMD by earlier detection of leukoplakia.

New management of patient care is necessary in order to improve the conditions for identifying preneoplastic lesions. Thus, a reminder and then a standardization of the recommendations must be made to dentists and hygienists during their training. Protocols must be followed by the greatest number of dentists in order to:✓Perform an annual check-up of the oral cavity of patients✓Address/refer or biopsy the detected mucosal lesions✓Ensure increased follow-up of drinker/smoker patients✓Ensure increased follow-up of patients who have already presented one or more cancers of the oral cavity

Regarding future dentists, the idea would be to target through questionnaires gaps in their training and to set up new procedures directly in universities during their Continuing Professional Education (lectures, clinical cases, strengthen links with maxillofacial surgeons). Recently, GEMUB (Groupe d’Etude de la MUqueuse Buccale, Study group of Oral Mucosa from Hôpital Trousseau, Chambray-lès-Tours, France) put in place recommendations for oral lichen (https://www.gemub.org/recommandations, accessed on 21 September 2020) [29].

These different points will make it possible to: Increase the detection rate of these lesions by the development of these clinical new standardized procedures. Oral cancers outcomes will be reduced only if the disease is recognized earlier.Change behaviors and lifestyle of the general population by encouraging people to go for a dental consultation. To contribute to health promotion, awareness campaigns, systematic reminders, promoting a free screening consultation for smokers/drinkers would be ways of approaching the public. A first option would be to improve the advice thus as not to scare patients, then to redirect them to a maxillofacial surgeon or an ENT surgeon if needed. Dentists must, therefore, be trained to initiate discussions (help smokers to give up their smoking habits and have good nutrition [9] and then in severe cases to announce bad news and address). A solution could be to set up key health messages as a “to-do dentist list” for oral cancer prevention. In the same line, dentists should pay particular attention to ethnic differences and lower socio-economically patients in the management of their patients, as they have a higher risk of developing OCSSC.Communicate: communication between the dentist and the patient provides information about him and the natural history of the lesion or call sign. Dentists might ask patients about the exams that they physically endure, making sure that the medical terminology often used is explained in layman language, talk about the future implications of patients as potential inclusion in clinical trials (understanding of notice formations, benefits/risks) [23].

### 3.2. The Patient’s Journey 

Patients with OPMD have a protracted healthcare route and may have multiple evaluations and interventions by a variety of medical caregivers. Some patients will be seen by a health practitioner several months up to years before and after the diagnosis of OCSCC, while others may be evaluated for a similar period without malignant transformation. After diagnosis or suspicion of cancer, the patient will be included in a care route through a multiple caregivers network. This network includes a number of specialists who will refer a patient to a cancer care pathway. An initial consultation will allow to quickly judge the advisability of the examination. Oral cancers can be treated in one hundred days, therefore, rapid management has to be performed. An endoscopic examination, a gastroscopy, and lesions burden analysis by imaging are necessary to guide the treatment, often administrated in combination. Clinical assessment and interventions often involve clinicians with heterogeneous training on oral cavity disorders (general practitioners, smoking-cessation specialists, coordinating nurse, assistants, orthoptics, dieticians, oncologists, dermatologists, dentists, ENT surgeons, maxillo-facial surgeons). Patient collaboration with all actors of the network is important to take into account, thus that patients are better informed upstream of their treatment.

### 3.3. The Help from Patient Advocacy Groups (PAGs) for a Patient-Centered Prevention

How can PAGs bring more awareness to preneoplasia preceding potential oral cancers and help patients after the identification of a suspicious lesion in the mouth? PAGs are today involved with awareness campaigns, lobbying, and education of both the health care system (HCS with HC personnel) as well as the survivor and the newly diagnosed. PAGs are a link between the clinician and the patient (Figure 2), making sure that the medical terminology often used is explained in layman language, that psychological support is available during and after treatment, and information beyond treatment options is explained. Active PAGs are included in most European-funded projects and have a role at the very center of the patient experience inclusion in terms of participation, engagement, involvement in research (particularly in a clinical trial). Regarding the prevention of OCSCC transformation into cancer, PAGs actions could be: ✓Raise creating screening campaigns✓Promote awareness of patient’s families and caregivers✓Develop surveys to map existing knowledge✓Raise awareness to use medical tools/devices for self-examination, for example, mirrors✓Create a national screening day✓Send routine dental checkups invitations (annual examination) with or without routine biopsy invitations, especially for drinkers/smokers✓Ask patients about the exams that they physically endure✓Encourage policies that eradicate exposure to risk factors

Therefore, the added value of patient representatives and researchers working together is the increased awareness of the problem [30] and at which angle to best attack it for better diagnostics, better treatment guidance, and ultimately better quality of life (Table 1).

## 4. A Fruitful Partnership

Patient associations increasingly support research on diseases they represent. Apart from contributing to the funding of scientific and clinical research, some of them actively participate in its orientation and in the production and dissemination of knowledge. There are various modalities of patient associations involvement in research activities. The influence of patient associations is related to the strategies developed by existing organizations by the public or private but also to the nature of the diseases they are fighting against. There is a crucial role of associations in the support to certain fields of research. Two types of roles based on the nature of the relationship associations establish with the professionals and of the role they play in the coproduction of knowledge. Through their involvement in research, some associations might contribute to the emergence of a new model for linking science users and science producers. This model might prove to be relevant in the future for other sectors (Figure 2).

Patient umbrella organizations have the possibility to receive feedback from member organizations through conducting large surveys that are disseminated throughout the national representations. The feedback received is used to map existing gaps in knowledge and provide an overview of the current problematic situation. With this, researchers can explore interesting areas where the research is lacking and focus the topics further. While being involved in the research projects as patient representations, the patient advocate can provide the aspects from a patient point of view as well as sharing the information and knowledge from the project with the organization it is representing. One example is the European Cancer Patient Coalition (ECPC) who are involved in many research projects both funded by the Europe commission as well as by privately funding companies. The network with health care professionals and researchers enables the organization to quickly be able to inform their member organizations regarding any medical terminology, relevant new clinical trials, the latest research publications, as well as recent implementations in the health care sector. By creating awareness campaigns such as the one presented in Figure 3 (https://makesensecampaign.eu/, accessed on 21–25 September 2020), patient organizations can make sure that the message is clear and easily digestible for their patient focus group and that the most relevant information is highlighted. The campaigns are being seen worldwide thanks to the different dissemination channels and the broad network between patient organizations that interact and collaborate with each other. Patient advocates frequently lobby and push policy- and decision-makers for bettering the currently existing policies and regulations. The campaigns being brought forward are often pushed to a member of parliaments (MEPs) and highlight the importance of increasing the budget or allocating funding to the chosen topic. Ultimately, research and health care professionals benefit from the collaboration with patient organizations since the patient advocates can help push the funding being allocated by government and politicians into areas that previously were not prioritized. 

### 4.1. Involvement in Research

Patient associations are often invited to take part in research projects to ensure the patient voice is included. This also enables the researchers to be aware of the priorities from a patient perspective and adapt their project to meet some of the unmet needs of patients. While sitting in advisory boards and participating in project meetings, patient representatives make sure to always include ethical aspects with patient needs in mind, as well as issues regarding data management that could affect GDPR regulations. The patient journey and some of the worries are often explained, and gaps of knowledge can be filled in where there previously was lacking due to not enough representations. The patient association usually takes over the responsibilities of disseminating the work conducted by the involved project, making sure that any crucially important patient information is reached to every patient on both a national and a European level. This solves a current problem that many researchers and consortiums have had, which is how to be able to reach the end person, which usually is the patients. A research funding proposal will have a heavier weight if patients are included, as the inclusion of all parties will enable better end products and minimize skewed opinions. These types of collaborations benefit both patients as well as research as the joint efforts can help focus funding received by the different private and public institutions. 

### 4.2. Advocacy

The strong backbone of all patient associations is their ability to advocate for their cause. By creating awareness campaigns, lobbying nationally as well as on a European level, patient advocates and patient representatives can bring awareness to the importance of prioritizing certain topics and/or areas of interest. In our modern life, politics, patient voices are becoming more and more important to have included in all aspects of decision making as the policymakers want to know where some of the main issues exist and what some of the key calls for actions are. By collaborating with patient associations, the awareness and focus that is brought to the specific project is, therefore, bigger and receives more attention. Events can be created in parliament together with MEP, and panel discussions can be held, webinars created, and social media campaigns that grasp the attention of the digital world globally. In the end, a unified voice can push more strongly than scattered echoes.

### 4.3. Extended Networks

Patient associations have an extended network of health care professionals, politicians, medical societies, and other organizations that they interact with on a regular basis. An umbrella patient organization as ECPC has over 500 member organizations who they can reach out to and ask for opinions and evaluations. Creating surveys to send out will reach a massive audience with just one click than any other institution will be able to reach. Newsletters with the latest information will be highlighted and brought forwarded to a large community. The outreach when partnering with patient associations is not only a huge benefit to all the involved partners but also a huge asset to the entire project team. The passion by patients to be involved in their own cause is overwhelming when you see the amount of effort, volunteering, and tired-less persistence each individual will bring to support its patient association and their specific topic. 

## 5. Conclusions

The involvement of patient associations in the research projects is only one part of a wider evolution, which has contributed to more than 50 years of reshaping relationships either, on one side, the specialists and professionals and, and on the other side, laymen or more precisely the groups concerned, who are the beneficiaries and recipients of this knowledge. This remodeling led simultaneously to patients to develop, among themselves, relationships of solidarity and support and to assert their rights to participate actively in the fight against diseases they are living with or have survived from. PAGs involvement takes varied forms, as we have noted. The emergence of new relationships between science and society is arising, and patients are at the center of this model, creating a new clinical model. 

In the field of preneoplasia, everything remains to be conducted: no awareness campaign has yet been carried out to monitor the development of these first symptoms, but PAGs as the European Cancer Patient Coalition are aware of the dissemination work needed to be performed and the importance of working hand in hand with patients for medical research, better prevention, and treatment advances. 

## Figures and Tables

**Figure 1 cancers-13-04408-f001:**
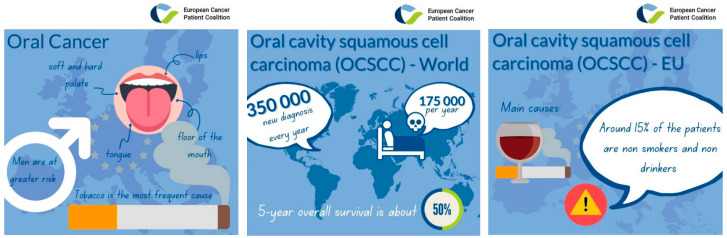
Oral cavity squamous cell carcinoma (OCSCC) incidence and main risk factors. 350,000 people are diagnosed with oral cavity squamous cell carcinoma (OCSCC) every year, and 175,000 patients die every year due to OCSCC. Tobacco is the most frequent cause, and only around 15% of the patients diagnosed with OCSCC in Europe are non-smokers and non-drinkers. Men are at greater risk than women, and the overall 5-year survival is about 50%.

**Figure 2 cancers-13-04408-f002:**
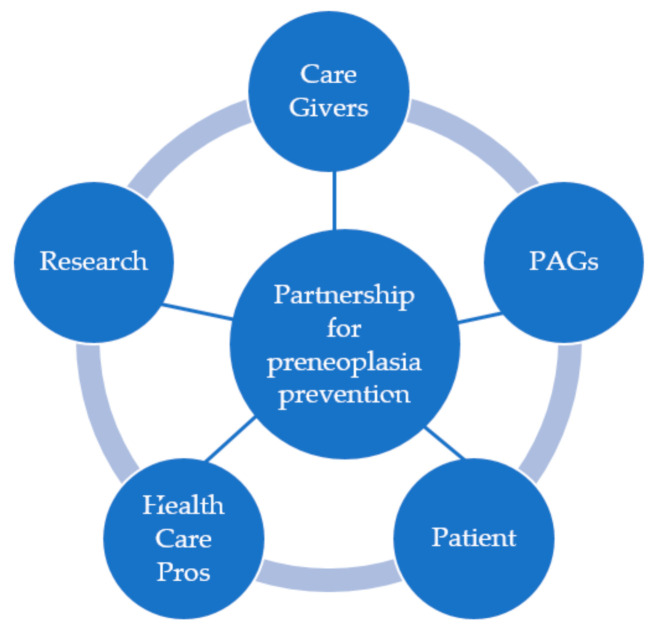
Scheme showing PAGs involvement. PAGs are today involved with awareness campaigns, lobbying, and education of both health care systems as well as the survivor and the newly diagnosed. The PAGs are a link between the clinician and the patient, making sure that the medical terminology used is explained in layman language and that psychological support is available during and after treatment.

**Figure 3 cancers-13-04408-f003:**
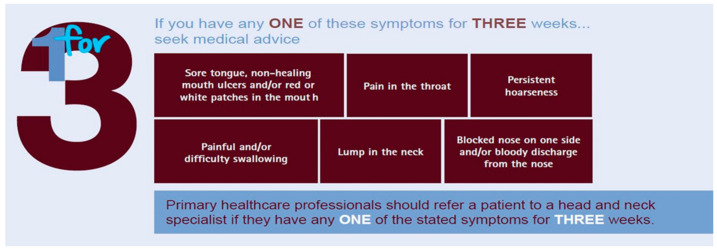
Awareness campaign example of ECPC. Head and neck cancer is a preventable and curable cancer, yet it continues to kill half of all sufferers. Awareness of the disease is alarmingly low; 73% of the general population in a 2020 pan-European survey were unaware of the disease’s symptoms. Early diagnosis and referral save lives. Therefore, there is an urgent need, not only to raise awareness of the signs and symptoms of the disease, but also to educate the general public and healthcare providers on the importance of prevention and regular screening.

**Table 1 cancers-13-04408-t001:** What PAGs are involved with currently, and what could be developed for other disease areas using similar campaigns.

Current Cancer PAGs Actions	Potential PAGs Preneoplasia Actions
Rising Awareness in H&N Cancers “Make Sense Campaign”	Rising Awareness in preneoplasia “Precancer Campaign”
Education on disease prevention	Screening campaigns with routine biopsies
Disease Understanding	Clinical Trials understanding to be sensitized
Signs and symptoms Understanding	Signs and symptoms education (tools/devices)
Encourage Early diagnosis	Encourage early diagnosis
Encourage formation to learn best practices	Importance of regular screening
Building partnerships with governmental bodies and organizations	Building partnerships with governmental bodies and organizations
Europe Standardizing Care	Europe Standardizing Care
Bring awareness to the importance of optimal facilities and expertise throughout the entire patient journey	Bring awareness to the importance of optimal, facilities and expertise throughout the entire patient journey

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
