# Peer review of "The 4P: Preventing Preneoplasia through Patients Partnership"

_cancers, 2021, doi:10.3390/cancers13174408_

Round 1

Reviewer 1 Report

The authors present a manuscript in which they deal with a very current and interesting topic, how can patient advocacy groups (PAGs) bring more awareness to preneoplasia preceding oral cancers and help patients after identifying a suspicious oral leukoplakia.

This reviewer believes that the authors should provide more data on initiatives similar to the one proposed, reporting the situation in France and other European countries.

This reviewer believes that the manuscript lacks data to support the thesis presented.

Furthermore, this reviewer believes that the work is more like a letter than a review in its current form.

In conclusion, this reviewer suggests to the authors either review their target by proposing to the journal the work in another form or enrich it with a detailed analysis of the available data that can support their thesis.

Reviewer 2 Report

In the manuscript " The 4P: Preventing Preneoplasia through Patients Partnership”, the authors aim to discuss the approach using patient advocacy groups to effectively prevent malignant transformation of oral potentially malignant disorders (OPMD).

There are some comments listed below:

  1. In Introduction, Page2 line 54, “Therefore, the main challenges for the prevention of leukoplakia transformation into cancer are to identify high-risk patients and treat them by chemotherapy in order to prevent OPMD transformation of the entire oral mucosa [5].”

-Since leukoplakia is only one of the OPMD, other lesions of OPMD can also transform to malignancy, even in higher transformation rate than leukoplakia, thus the “leukoplakia” here is better changed to OPMD.  

  1. In introduction, page 2-3 line 84-98, “2.1. World Health Organization (WHO) distinguishes 3 types of prevention [8]”: primary, secondary, tertiary…

-The arrangement of 2.2 secondary prevention… is better to be arranged as primary and tertiary.

  1. In introduction, page 3 line 112, “2.3. Primary Prevention: No smoking/No drinking campaigns”

   -Because subtitle for primary prevention has appeared in paragraph above, it is slightly confusing. Modify the subtitle will be better.

   -The contents in primary prevention (second and third paragraph) should be placed in secondary prevention.
